# Liposomes as Delivery System for Applications in Meat Products

**DOI:** 10.3390/foods11193017

**Published:** 2022-09-28

**Authors:** Li Huang, Wendi Teng, Jinxuan Cao, Jinpeng Wang

**Affiliations:** 1School of Food and Health, Beijing Technology and Business University, Beijing 100048, China; 2College of Food and Biological Engineering, Chengdu University, Chengdu 610106, China

**Keywords:** liposome, meat products, applications, antibacterial, antioxidant

## Abstract

In the meat industry, microbial contamination, and lipid and protein oxidation are important factors for quality deterioration. Although natural preservatives have been widely used in various meat products, their biological activities are often reduced due to their volatility, instability, and easy degradation. Liposomes as an amphiphilic delivery system can be used to encapsulate food active compounds, which can improve their stability, promote antibacterial and antioxidant effects and further extend the shelf life of meat products. In this review, we mainly introduce liposomes and methods of their preparation including conventional and advanced techniques. Meanwhile, the main current applications of liposomes and biopolymer-liposome hybrid systems in meat preservation are presented.

## 1. Introduction

Liposomes are self-assembled spherical bilayer delivery systems with amphiphilic properties, which develop by phospholipids with an aqueous ingredient inside and a hydrophobic layer on the surface [1]. Because of their biocompatibility, biodegradability, nontoxicity, and non-immunogenicity, liposomes are considered as powerful drug delivery systems [2]. The continuous release and stability of substances can be achieved by liposomal encapsulation which is mainly applicated in food science, cosmetics, and medical fields.

Meats are a compact matrix with excellent viscoelastic and structural properties, which contain high contents of protein, saturated or unsaturated fat, water, and vitamins. They can be further processed into sausage, bacon, burger, dumpling, or ready-to-eat food. However, during processing and storage, meats are inclined to suffer from lipid and protein oxidation, resulting in the production of hydrogen peroxide, volatile compounds, and genotoxic amino acid derivatives, adversely affecting the digestibility, availability, and sensory of meat products [3]. Furthermore, the abundant nutrients in meats make it an ideal seminary for contamination and proliferation of microorganisms, leading to food decay and deterioration. In this context, synthetic substances like propyl gallate (PG), tert-butylhydroquinone (TBHQ), butylated hydroxyanisole (BHA), butylated hydroxytoluene (BHT) are largely applied to preserve the quality of meats, but consumers prefer natural antioxidants since they quite worry about potential toxicological effects of synthetic ones [4]. Therefore, natural herbs, spices, plant extracts and plant essential oils, peptides, chitosan, bacteriophages can all be used as potential alternatives to instead of synthetic antimicrobial/antioxidant agents to improve the quality and safety of meat products [5]. However, the applications of these natural preservatives are limited due to their pungent odor, volatility, and instability. To overcome these shortcomings, liposomal encapsulation can be applied to the meat industry to preserve and release these components.

Much public research and reviews have been published on the applications of liposomes as bioactive compounds delivery systems in food [6,7,8,9]. However, there is no comprehensive review on the applications of liposomes in meat products. This review first introduced the preparation of liposomes, including conventional and advanced technologies. Then, the applications of antimicrobials/ antioxidants-loaded liposomes in meat products were discussed in detail. Besides, the integration of liposomes and biopolymers to prepare biopolymer-liposome hybrid systems for meat preservation was also summarized. Finally, the gaps in current research were emphasized to indicate the future research directions and prospects in meat products.

## 2. General Information of Liposomes

Liposomes are spherical, closed structures, composed of curved lipid bilayers, which enclose part of the surrounding solvent in their interior [10]. Hydrophilic compounds in the aqueous cavity, hydrophobic compounds within the lipidic membrane as well as amphiphilic substances can be incorporated within these vesicles (Figure 1A) [11]. Phospholipids are the main components of liposomes, which are amphiphilic lipids composed of a hydrophilic polar head group and two hydrophobic fatty acid chains. The most widely used natural phospholipids are soy lecithin, egg lecithin [12], marine phospholipid [13], and milk phospholipid [14]. Synthetic phospholipids are produced by modifying the head group, fat chain, and alcohol of natural phospholipids, which are more stable than the natural phospholipids [15], such as 1,2-dis-tearoyl-sn-glycero-3-phosphocholine (DSPC), 1,2-dipalmitoyl-sn-glyc-ero-3-phosphocholine (DPPC) and so on [16]. Cholesterol is also an important component of liposomes, which is composed of hydroxyl, four-ring steroid skeleton, and hydrocarbon tail. It cannot form vesicles by itself, but it can be incorporated into the phospholipid membrane in very high concentrations of up to a 1:1, or even a 2:1, molar ratio of cholesterol to phosphatidylcholine. In membranes, its aliphatic chain is aligned parallel to the acyl chains in the center of the bilayer, and the hydroxyl group is oriented towards the aqueous surface (Figure 1B) [17]. The incorporation of cholesterol can influence lipid bilayer fluidity, reduce their permeability, and increase their in vitro and in vivo stability. Besides, there are still many components that can enhance the stability of liposomes, including propylene glycol and polyethylene glycol (PEG), phytosterols (PS)/phytosterol esters (PEs) [18], sea cucumber sulfated sterols [19], gum arabic (GA)/sodium alginate (SA) [20], guar gum (GG) [21], pectin [22], even chitosan [23] as coating materials.

Liposomes can be classified as neutral, negative, and positive liposomes based on their surface charges. Besides, according to their structures, liposomes can be classified on the number of lipid bilayers (lamellae) or vesicle sizes (Figure 2). Based on their lamellarity, liposomes can be divided into unilamellar (ULV, all size range), multilamellar (MLV, >500 nm) and multivesicular (MVV, >1000 nm) vesicles. Because of their different sizes, ULV can be either a small unilamellar vesicle (SUV, 20–100 nm), a large unilamellar vesicle (LUV, >100 nm), or a giant unilamellar vesicle (GUV, >1000 nm). ULVs have the ability to encapsulate hydrophilic compounds with the presence of a single bilayer. MLVs present two or more concentric lipid bilayers organized by an onion-like structure, favorably for the encapsulation of lipophilic compounds. MVVs include several small non-concentric vesicles entrapped within a single lipid bilayer, which are very suitable for encapsulating large volumes of hydrophilic ingredients [2]. The vesicle size is a vital criterion to determine the circulation half-life of liposomes, and both the number of bilayers and size will influence the loading capacity of active components [7].

## 3. Preparation of Liposomes

In the past few years, various kinds of techniques have been developed for liposome preparation (Table 1). Different techniques would influence the physicochemical properties of liposomes, such as size, lamellarity, encapsulation efficiency, and so on. They could be categorized as conventional and advanced methods, as well as size reduction methods, which were discussed and explored in the following section.

### 3.1. Conventional Methods

There are a wide variety of conventional techniques used for liposomal preparation. Lipids combined with an aqueous phase are needed for all conventional methods. 

Thin film hydration is the most commonly used method for liposome preparation. In this method, lipids and amphiphilic molecules are first solubilized in an organic phase. Then, the solvent is evaporated by using a rotary evaporator under a vacuum, leaving a thin film of lipids. After the thin film is hydrated by an aqueous solution, a liposomal is formed. Relying on different hydration conditions, liposomes with different structures are developed. An intense shaking during the hydration generates MLVs with heterogeneous sizes, while mild hydration produces GUVs [1,2]. This method is widely used and easy to handle in the lab. However, there are still various problems, such as low entrapment ability, and hard complete removal of organic solvent, which limit its applications on an industrial scale.

Reverse phase evaporation helps form a mixture of LUVs and MLVs. In this method, lipids and amphiphilic molecules are first mixed and dissolved in an organic solvent. Then, an aqueous buffer containing a solubilized active compound is added to the mixture. Finally, the organic solvent is evaporated under reduced pressure, remaining liposomes in the aqueous media [1]. This method, on the one hand, provides a high EE. On the other hand, since the loaded compounds would make contact with an organic solvent, it is not suitable for fragile molecules like peptides [10]. 

Solvent injection method involves the dissolution of lipids into an organic phase (ethanol or ether), followed by the rapid injection of the lipid solution into aqueous media, forming liposomes [11]. The ethanol injection method is rapid, simple, and reproducible. At the same time, it does not cause lipid degradation or oxidative alterations. However, the insolubility of some lipids in ethanol, heterogeneous sizes of liposomes, very low EE of hydrophilic compounds, and incomplete removal of organic solutions are common concerns [2]. Other than the ethanol injection, since ether and water cannot dissolve each other, heating is always used to remove ether from the formed liposome [10]. 

The detergent removal method is another known technique to produce liposomes. It is a mild process for producing MLVs and LUVs. Based on the construction of detergent-lipid micelles, the detergent is removed to form liposomes. The disadvantages of this method are that the concentration of liposomes and EE of loaded compounds are quite low. The size and homogeneity of liposomes are based on the rate of detergent removal and the initial ratio of detergent to lipid. Besides, it is very time-consuming and often remains the detergents [10]. 

For the emulsion method, phospholipids are dissolved into an organic phase and mixed with an aqueous phase to form a water-in-oil (W/O) emulsion. The mixture is then added to another aqueous medium to form a double emulsion (W/O/W). The organic solvent is then evaporated, forming liposomal. This method does not work continuously, and the amount of production is limited. MVVs can be obtained in this way. But complete removal of organic solvent needs to be conformed [28]. 

The heating method was developed by Mozafari [30]. Phospholipids are dissolved into aqueous media with 3% *v*/*v* glycerol at a high temperature. Glycerol is used because of its water solubility, isotonic, and ability to increase the stability of lipid vesicles and prevent coagulation and sedimentation [30]. The heating method does not involve organic solvents or detergents [31]. High temperature, on the one hand, induces minor degradation of bioactive lipids, on the other hand, avoids further sterilization, thus decreasing time consumption and process complexity [7].

Preparation of liposomes of homogeneous size is quite necessary for applications in the fields of food, medicine, and material science since it could influence the physiochemical properties such as stability, EE, release, and cellular uptake [25] In general, the smaller the particle size, the more stable and uniform it is. There is a lower tendency for aggregation and physical instability during storage [32]. Reducing the particle size is helpful to increase the diffusion ability of the carrier and facilitate uniform distribution in vitro. In comparison with liposomes, nanoliposomes provide more surface area and circulate in the organism for a long time [2,33]. Furthermore, tissue distribution and clearance of liposomes in vivo are related to particle size [33]. Therefore, liposomes produced by traditional methods still require additional techniques, such as homogenization, extrusion, and sonication (probe sonication and bath sonication), to help reduce their sizes [12]. For homogenization techniques, liposomes would be forced to pass within an orifice under high pressure to reduce the sizes by a high-velocity collision. Microfluidization, high-pressure homogenization, and shear can be included in this category of size reduction. In the extrusion process, the liposomes would pass through a membrane of a defined pore several times to generate uniform size distribution. Compared with homogenization, this process needs much lower pressure and less volume of liposomal suspension [2]. Sonication is most extensively used for the preparation of SUVs, whose main disadvantages are very low internal volume/EE, possible degradation of phospholipids and loaded compounds, and metal pollution from the probe tip [12].

### 3.2. Advanced Methods

Conventional methods have various shortcomings, such as poor mono-dispersion, poor stability, high residual organic solvent, and many other side effects. Therefore, advanced methods have been developed to overcome the above problems and accelerate the scale-up of industrial production. Some of the advanced methods are based on the modification or improvement of conventional methods. For example, cross-flow filtration improved the detergent removal method [34]. Both cross-flow injection and membrane contractor technology modified the ethanol injection method [11]. Others, like dual asymmetric centrifugation, microfluidics production, and supercritical fluid (SCF) methods have great potentials to be developed as alternatives for liposome production.

Dual asymmetric centrifugation (DAC) is a special kind of centrifugation, where the tube is turned around its center (vertical axis) at a specific distance and speed. The energy produced by mechanical turbulence and cavitations forms nanoliposomes with homogenous size distribution. DAC has a high trapping efficiency, but it is used only for small volumes on a laboratory scale [10].

The microfluidic method is a promising technique for controlling the mixing of fluid streams in a microfluidic channel. Accurately dispensed nanoliter volumes, exact control over the interface, diffusion-dominated axial mixing, and continuous mode of operation at low volumes are their advantages [26,27], whereas sophisticated devices remain expensive and time-consuming. Also, volumetric throughput has been greatly limited in view of the traditional layout of microchannels. Recently, a 3D-printed integrated microfluidic chip was developed to prepare ultra-high volumetric throughput nanoliposome and could control the size efficiently [35].

Supercritical fluids (SCFs) integrate some satisfying properties of both liquids and gases, a small change in pressure or temperature resulting in a large change in the density of SCF or the solubility of compounds in the SCF [26]. Researchers are increasingly replacing organic solvents with SCFs. The most extensively used supercritical gas is carbon dioxide (CO_2_), due to its non-flammable, low-toxic, cheap, environment-friendly, and easily manageable properties [25]. Compared with the traditional method, SC CO_2_ has promoted the intactness, sphericity, and homogeneity of liposomes, which enables phospholipids to aggregate into nano/microparticles by regulating the rate of decompression and the opening diameter of the nozzle [25]. However, the high cost and pressure of sophisticated instruments limit their applications [7].

Overall, both conventional and advanced techniques can be utilized for liposome preparation, according to different conditions and needs. In the production of liposomes, EE, stability, particle size and distribution, residual organic solvents and other factors need to be considered. Moreover, large-scale and continuous industrial production still faces great challenges.

## 4. Application of Liposomes in Meat Products

The applications of liposomes in meat products have been extensively explored mainly in three aspects: physicochemical properties including controlled release and improved bioavailability, meat preservation by encapsulating natural antimicrobials and antioxidants, and integration of biopolymer-liposome hybrid systems including single/multilayer coating liposomes, active edible films and electrospinning nanofibrous membranes (Figure 3).

### 4.1. The Release and Bioavailability of Delivered Molecules

Natural active compounds are often easily volatilized and rapidly degraded so that their action period is shortened, and the action effect is weakened due to the harsh environments. Sustained release systems are capable of achieving a prolonged effect by slowly releasing active compounds over an extended duration [39,40]. Researchers used liposomes to encapsulate laurel essential oil and silver nanoparticles to control the pungent flavor of essential oil and reduce the toxicity of silver nanoparticles [36]. Chitosan and pectin-coated chrysanthemum essential oil liposomes significantly slowed down the release of chrysanthemum essential oil and obtained a continuous antibacterial effect against *Campylobacter jejuni* on chicken [41]. In addition, stimuli-sensitive liposomes have been developed, which depended on different environmental factors, such as pH, magnetism, temperature, enzymes, and so on, to trigger the release of the bioactive molecule [42]. For instance, the basil essential oil cationic liposomes nanofibrous system was engineered with responsiveness to phospholipase secreted by *Listeria monocytogenes* [43]. Taking advantage of a bacterial protease secreted from *Bacillus cereus*, the controlled release of cinnamon essential oil from proteoliposomes was achieved via proteolysis of protein in proteoliposomes [44].

Bioavailability is defined as the relative amount of an active ingredient that is absorbed into the bloodstream after ingestion by the body [45,46]. When the liposomes are hydrolyzed and then reconstructed to form mixed micelles of phospholipids, bile salt, and fatty acids, the poorly water-soluble compounds would be transferred and solubilized into the micelles' hydrophobic core, resulting in increased solubilization of the hydrophobic compound under physiological conditions [6,47]. Then, the bioavailability of loaded compounds could be improved. For example, ferric pyrophosphate showed low bioavailability because of its poor water solubility [48]. It had been studied that iron pyrophosphate was encapsulated in liposomes to improve the bioavailability of iron, and further added to meat pate as a fortifier to prevent iron deficiency in humans [49,50].

### 4.2. The Antibacterial and Antioxidant Effects

Foodborne pathogens and oxidation are two dominant factors that influence the quality of meat products during their preparation and storage [41]. To keep meat fresh and extend its storage period, functional antibacterial and antioxidant agents are attracted much attention. Liposomes as a type of active food carrier have been applied in meat preservation. In this section, the antioxidant and antibacterial mechanisms of meat preservation were illustrated, as well as the applications of encapsulated natural preservatives in the meat industry were discussed.

#### 4.2.1. The Antioxidant and Antibacterial Mechanisms in Food Products

(1)The antioxidant mechanism in food products

Lipid oxidation is responsible for the loss of nutritional quality and changes in the flavor and color of meat, resulting in health hazards and economic losses. It is a highly complex set of free radical reactions between fatty acids and oxygen. Under thermal, redox, or light stimulus, prooxidants, reactive oxygen species (ROS) and any other oxidation-favorable conditions contribute to the loss of a hydrogen radical from fatty acids. Then, oxygen reacts with the alkyl radical of fatty acid and results in peroxide radical formation, followed by hydroperoxides, the primary products of lipid oxidation form. When harsh conditions are presented in the meat, the hydroperoxides become vulnerable to further free radical chain reactions, such as isomerization or decomposition, producing secondary products including pentanal, hexanal, and malondialdehyde. Finally, the free radicals react to form stable products [51,52]. Protein oxidation refers to the covalent modification caused by the direct reaction of protein with ROS or indirect reaction with oxidative stress by-products. Similar to lipid oxidation, protein oxidation in meat is carried out through the chain reaction of free radicals [53]. Natural antioxidants such as phenolic compounds, flavonoids, vitamins, and peptides could direct or indirectly inhibit and delay oxidation mainly by absorbing UV radiation, quenching singlet oxygen, chelating metal ions (Fe^2+^, Fe^3+^ and Cu^2+^), scavenging free radicals, decomposing hydroperoxides and inhibiting enzymes (Figure 4) [54,55,56].

(2)The antibacterial mechanism in food products

Meat products include a high abundance of proteins, lipids, water, and vitamins, which are suitable for microbial growth and reproduction. Natural antimicrobial agents can act on microbial cells through the following mechanisms (Figure 5): (i) interact with various components of the bacterial cell wall (such as peptidoglycan and lipopolysaccharide) to destroy the cell wall (ii) interact with cell membranes to disrupt the phospholipids or lipid bilayers, further affecting the fluidity and permeability of the membrane, leading to the formation of pores and leakage of intracellular components; (iii) damage DNA and RNA and interfere with protein synthesis; (iv) inhibit the activity of various enzymes needed for bacterial growth, such as DNA gyrase, fatty acid synthase, ATP synthetase and so on; (v) interrupt electron transport process to influence respiration and energy metabolism, inhibit the secretion of bacterial toxins [57,58,59,60].

#### 4.2.2. Applications of Encapsulated Active Compounds in the Meat Industry

Liposome acts as a protective barrier against environmental factors to protect and stabilize the encapsulated compounds. It is well documented that liposomes can improve stability (light, heat, pH, and so on). For example, the starting decomposition temperature of astaxanthin in nanoliposomes was higher than that of free astaxanthin, indicating that encapsulation could effectively enhance the thermal stability of astaxanthin [61]. The degradation of curcumin in nanoliposomes was significantly lower than that of free curcumin under alkaline conditions [62]. In the meat industry, to protect from microbial contamination and lipids and protein oxidation, liposome technology has been applied to load antimicrobial and antioxidant agents such as essential oils, bacteriophages, peptides, and bioactive compounds (Table 2).

(1)Essential oils

Essential oils can act as preservative agents and active packaging ingredients for foods. Despite all these applications, less stability, heat sensitivity, and the pungent odor of essential oils are the main constraint to their effective exploitation. Liposome has been an efficient tool to overcome these drawbacks. For example, nutmeg essential oil [63,64,65], thyme essential oil [37], and *Zataria multiflora* Boiss. essential oil [66] were incorporated into the liposomes system to enhance stability and extend active time. Nutmeg oil, as a main functional component of nutmeg, was often used in meat products because of its good antibacterial and antioxidant properties [75,76]. But studies showed that nutmeg oil was much more volatile, and its antibacterial effect was weakened after 5 days so the number of bacteria had reached 8.3 Log CFU/mL again. Interestingly, when nutmeg oil was loaded with liposomes, the stability and antibacterial effect were significantly enhanced. After 7 days of storage, the number of bacteria remained at around 5.0 Log CFU/mL, with a 99.9% of sterilization rate [63]. Another research showed that nutmeg oil encapsulated in liposomes could extend the antimicrobial time and effect on *L. monocytogenes* in dumplings [64]. In addition, nutmeg essential oil-encapsulated solid liposomes not only inhibited the growth of microorganisms, but also delayed the proteins and lipids oxidation in pork meat batters, which modified the viscoelasticity, interfered with the aggregated process of meat proteins, as well as greatly preserved the texture and color of pork meat batters [65]. Similarly, compared with free thyme oil, the thyme oil-encapsulated liposome could significantly improve the bacteriostatic effects against *S. enteritidis* without causing any health risk and negative sensorial impact in chicken [37]. *Zataria multiflora* Boiss. the essential oil in the nano-liposomal form increased the antioxidant and antimicrobial activity in beef burgers at 4 °C [66].

(2)Bacteriophages

Bacteriophages are viruses that are capable of invading bacterial cells. Lytic bacteriophages disrupt bacterial metabolism leading to the death of the bacteria [5]. The use of bacteriophages in food for the biocontrol of pathogens has fostered much attention, but they are susceptible to losing bioactivity in food due to the presence of enzymes, acidic conditions, and others [77]. To enhance their stability, liposomes are introduced as effective delivery vehicles to protect phages from harsh environments. Lin, Zhu, and Cui [67] fabricated a novel poly-L-lysine-coated bacteriophages liposomes, which markedly enhanced the stability and antibacterial effect on *E. coli* O157:H7, but with no impact on the quality of the pork. Cui, Yuan, and Lin [68] showed that chitosan film embedded with liposome-loaded phage exhibited high antibacterial activity against *E. coli* O157:H7 without changes in the sensory properties of beef.

(3)Bioactive compounds

Many bioactive compounds derived from plant extracts, such as polyphenols, flavonoids, organic acids, and tannins have excellent antibacterial and antioxidant activities. But their volatilization and oxidation sensitivity may lead to a decrease in functional activity. Therefore, it is necessary to use liposomes to achieve high activity. The extract of bay leaf (*Laurus nobilis*) had abundant phenolic and flavonoid compounds. Studies showed 1500 ppm bay leaf extract-loaded nanoliposome in the minced beef not only significantly delayed the process of microbial spoilage and oxidation, but also inhibited the growth of *Staphylococcus aureus* and *Escherichia coli*, showing that bioactive compounds-encapsulated liposome could be used to extend the shelf life of beef without causing undesirable odor and taste, in terms of low oxidative stability and microbial spoilage of free extract [69]. Lupulon and xanthohumol, as two acidic components derived from hop extracts, not only can eliminate the bitterness of beer, but also have antimicrobial and antioxidant activity [78]. Lupulon–xanthohumol loaded nanoliposome could be used as natural bioactive additives to replace partial nitrite in cooked beef sausage (60% meat), which remained microbiologically safe and postponed the oxidation during 30-day storage at 4 °C. Surprisingly, it did not impair the sensory properties of the final product [70,71]. Catechin is a polyphenol with synergistic antimicrobial and antioxidant properties that can be applied to meat products. Compared with free catechin, liposome technology significantly enhanced the antioxidant and antibacterial effects of catechin in Chinese dried pork during 25 days of storage at room temperature [38], effectively inhibited bacterial reproduction, extended the shelf life of sauce duck to more than 24 days [72], and increased the nitrosamines inhibition in traditional Chinese bacon after long-term storage [73]. 

(4)Peptides

Antibacterial peptides and antioxidant peptides are easily absorbed by the human body and have certain biological activities, which are widely used to prevent food oxidative deterioration and microbial degradation. Research showed that encapsulated peptides retained 87% initial antioxidant activities after 30 days, which was higher than free peptides [79]. In addition, antimicrobial peptide nisin entrapped liposomes could retain between 70–90% EE despite exposure to elevated temperature and alkaline or acidic pH [80] These results suggested that liposomes can protect peptides from harsh environments [81]. In meat products, quinoa peptide-loaded liposomes inhibited the growth of microorganisms (total bacterial count, *Staphylococcus aureus*, mold, and yeast counts), reduced primary (peroxide values) and secondary (TBARS values) lipid oxidation as well as the decomposition of protein (TVB-N) in burgers. Compared to free peptides, their antimicrobial and antioxidant properties in burgers were improved [74].

### 4.3. The Biopolymer-Liposome Hybrid Systems

Conventional liposomes are prone to aggregate, fuse, degrade or hydrolyze, resulting in the leakage of entrapped compounds. At the same time, their shortcomings of low resistance to gastrointestinal environments, poor ability to respond to external stimuli, and rapid elimination from blood circulation still need to be solved [82]. Integration of liposomes and biopolymers provides a promising strategy for overcoming these limitations related to stability. Herein, we are primarily concerned with a variety of biopolymer-liposome hybrid systems used for meat preservation in recent years (Table 3).

Biopolymer-coated liposomes are extensively studied because of their simple and efficient preparation. The single-layered liposomes can be obtained by directly mixing a polymer solution and a liposome suspension. Their stabilizing effects are mainly attributed to the enhanced electrostatic and steric repulsion provided by the coated layer [82]. Therefore, various biopolymers, such as chitosan and its derivatives, cellulose, pectin, polysaccharide gums, and proteins can be applied to liposomes as coating materials. Chitosan has a linear cationic structure of (1→4)-linked glucosamine and N-acetyl-glucosamine that can be obtained by partial deacetylation of chitin. Based on the electrostatic interaction between the protonated amino groups of chitosan and negatively charged groups presented on the surface of liposomes, chitosan-coated liposomes could be formed [89]. Taking *Satureja khuzestanica* essential oils (SKEO) as an example, chitosan coating with nano-liposomal SKEO showed a much better antibacterial and antioxidant effect on lamb meat [83]. On the one hand, chitosan layer formed physical barriers to prevent vesicle aggregation and improve the stability of the liposomes. On the other hand, it lowered the diffusion of oxygen on the meat surface, consequently inhibiting lipid oxidation by providing dense mesh covering. Chitosan-coated liposomes can also be used to load multicomponent as co-delivery systems. For example, laurel essential oil was encapsulated in the lipid bilayer by hydrophobic interaction, and silver nanoparticles treated with lignin were encapsulated in the internal aqueous core by hydrophilic interaction (Figure 6A). The prepared chitosan-coated liposomes effectively maintained the quality of pork at 4 °C for 15 days owed [36].

Apart from single-layered coatings, multilayered layered coatings were also developed to enhance the protective efficacy of liposomes. To improve the stability of liposomes, chitosan with a positive charge was used as the second layer to coat chrysanthemum essential oil (CHEO)-loaded liposomes with a negative charge, followed by pectin with a negative charge as the third layer (Figure 6B). Results revealed that CHEO triple-layer liposomes were more stable than single-layer and double-layer liposomes. Besides, CHEO triple-layer liposomes possessed higher antibacterial activity against *Campylobacter jejuni* on chicken during 14-day storage at a temperature range from 4 to 37 °C, without impact on the quality of chicken [41].

Biopolymer-based edible films have been widely exploited as active packaging systems. Chitosan, as a material for edible films, has excellent film-forming properties, broad antibacterial performance, and compatibility with other substances [90]. The incorporation of antimicrobial agents-loaded liposomal into chitosan films can be applied to extend the shelf life of food products. For example, active edible films based on chitosan loaded with nano-liposomal garlic essential oil (Figure 6C) presented significant synergistic antibacterial and antioxidant effects on chicken fillets [84]. Besides, active edible films, based on whey protein or chitosan incorporated with nano-encapsulated garlic essential oil (NGEO), retarded lipid oxidation and the growth of main spoilage bacterial groups in cooked sausages [85]. Gelatin is also used to produce active films with a high film-forming ability [91]. Betanin-loaded nanoliposomes incorporated gelatin/chitosan nanofiber/ZnO nanoparticles nanocomposite film was fabricated to package fresh beef. Research confirmed it could control the growth of bacteria, lipid oxidation, and the changes in the pH and color quality of the beef [88].

Electrospinning is an emerging technique to produce continuous polymeric fibers with the advantages of simplicity, economy, flexibility, and scale for mass production. The prepared functionalized nanofibers have the characteristics of small diameters, high surface-to-volume ratio, suitable porosity, and high encapsulation efficiency for bioactive compounds [92]. The biopolymer-based nanofibers showed significant potential in the encapsulation of bioactive ingredients and food packaging coatings. For example, a novel antibacterial packaging material was engineered by incorporating cinnamon essential oil/β-cyclodextrin (CEO/β-CD) proteoliposomes into poly (ethylene oxide) (PEO) nanofibers by electrospinning technique (Figure 6D). After the treatment of CEO/β-CD proteoliposomes nanofibers packaging, the satisfactory antibacterial efficiency against *B. cereus* on beef was realized without any impact on sensory quality of beef [44]. Besides, SiO_2_-eugenol liposome-loaded electrospun nanofibrous membranes exhibited excellent antioxidant activity on beef [86]. Tea tree oil liposomes/chitosan nanofibers could prevent microbial contamination by *Salmonella* to extend the shelf life of chicken meat [87]. In a similar way, basil essential oil-loaded cationic liposomes were incorporated into polyethylene oxide/soybean lecithin-based nanofibers (BCL-NFs). Results showed that the prepared antibacterial nanofibrous mats could help maintain the quality of chilled pork during 4-day storage [43].

## 5. Conclusions and Perspectives

Liposomes can be widely applied in the meat industry to protect sensitive components from degradation, increase the bioavailability of micronutrients like iron, promote the efficacy of food additives, such as antimicrobial and antioxidant agents and extend the shelf life of meat. In addition, liposomes could be combined with polymer coating (chitosan, pectin), nanoparticles (AgNPs, CuNPs, ZnO NPs), or cyclodextrin to fabricate composite biological films or nanofibers for the preservation of meat products. They not only improve the stability of liposomes, but also show synergistic antibacterial and antioxidant potentials as active food packaging materials. This review can help understand the liposome systems, as well as provides a scientific basis for the application of liposomes in meat products.

Encouraged by the vigorous development of liposomes, we have proposed several research perspectives in both basic research and applications: (1) A variety of multi-component delivery systems could be designed to load hydrophilic and hydrophobic bioactive ingredients simultaneously in liposomes. For instance, the co-encapsulation of vitamin C and *β*-carotene synergistically enhanced the antioxidative activity [93]. (2) New, generally recognized as safe (GRAS) materials composed of natural biopolymers should be explored for the preservation of meat products. (3) The controlled release of substances at specific sites or environments should be realized, which still needs to be investigated for the development of stimuli-responsive liposome systems. (4) It is worth noting that not all liposome applications can bring a beneficial effect. For instance, marine fish skin peptide-loaded liposomes were added to pork patty, and it showed serious fat oxidation [94]. Compared to free grape seed extract, liposomal encapsulation did not significantly improve the inhibitory effect on heterocyclic aromatic amine formation during frying beef patties [95]. These phenomena may be due to the structures and physicochemical properties of the liposome encapsulation system and its interaction with complex food matrices. However, comprehensive information on the interaction between liposomes and ingredients in the food system is scarcely discussed. Future work on the application of liposomes in food science should focus on their properties entering the food system and their interactions with complex food matrices.

## Figures and Tables

**Figure 1 foods-11-03017-f001:**
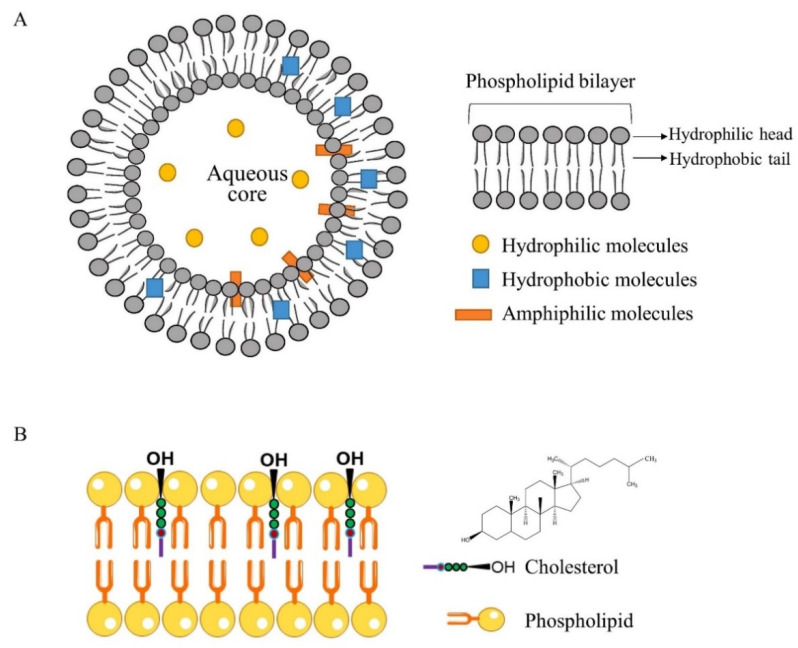
(**A**) The general structure of liposomes. Reprinted with permission from [2]. Copyright (2021), Elsevier. (**B**) The location of cholesterol in the phospholipid bilayer membrane. Reprinted with permission from [24]. Copyright (2019), Elsevier.

**Figure 2 foods-11-03017-f002:**
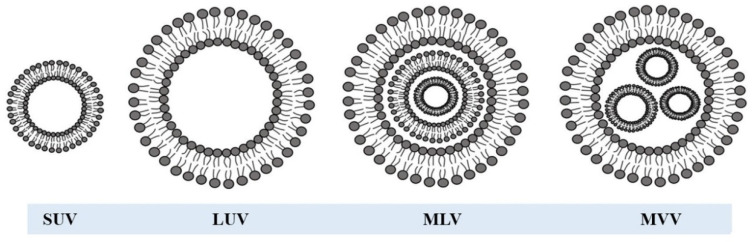
Liposomal classification is based on lamellarity and size. Reprinted with permission from [2]. Copyright (2021), Elsevier. SUV: small unilamellar vesicle; LUV: large unilamellar vesicle; MLV: multilamellar vesicle; MVV: multivesicular vesicle.

**Figure 3 foods-11-03017-f003:**
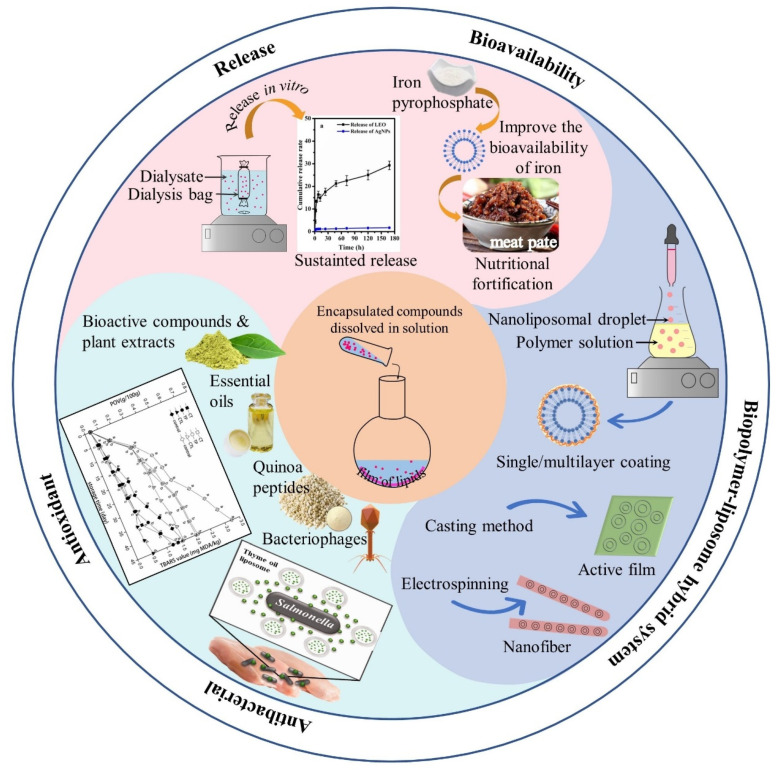
Overview of the applications of liposomes in meat products. (Adopted from Wu, et al. [36]; Cui, et al. [37]; Wu, et al. [38]).

**Figure 4 foods-11-03017-f004:**
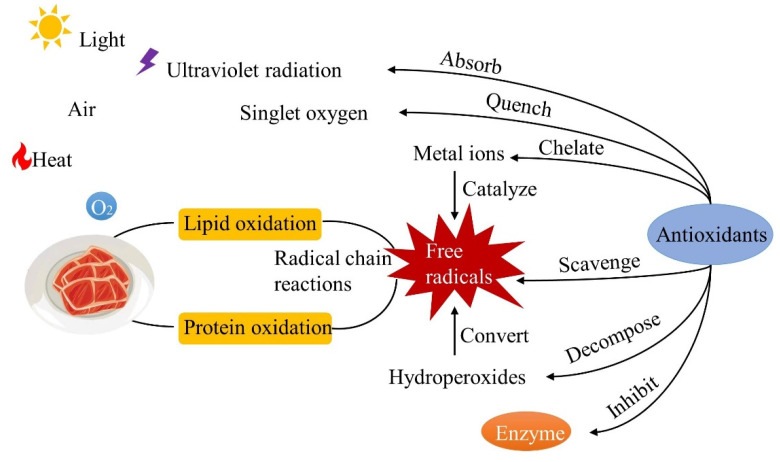
Antioxidant mechanism on meat preservation.

**Figure 5 foods-11-03017-f005:**
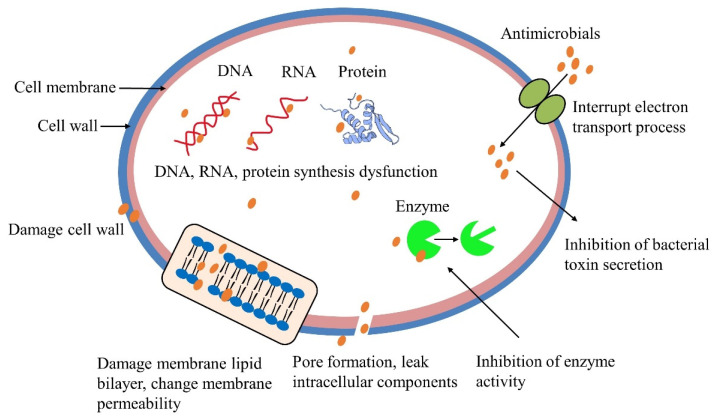
Antibacterial mechanism on microbial cells.

**Figure 6 foods-11-03017-f006:**
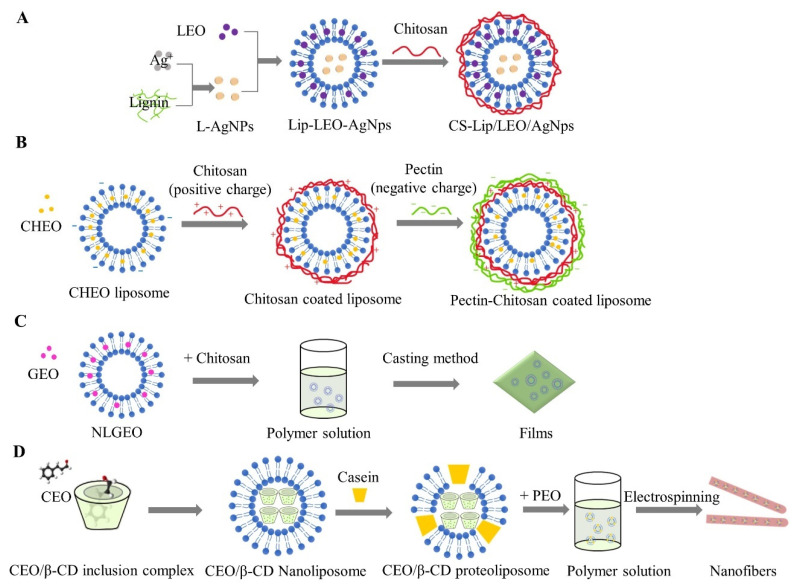
(**A**) Schematic preparation of CS-Lip/LEO/AgNPs for pork preservation. (Modified from Wu, et al. [36]). (**B**) Schematic preparation of pectin-chitosan coated liposomes (**C**) Schematic preparation of NLGEO incorporated into chitosan films. (**D**) Schematic of the preparation of CEO/β-CD proteoliposomes incorporated into PEO nanofibers. (Modified from Lin, et al. [44]). LEO: laurel essential oil; AgNPs: silver nanoparticles; CS: chitosan; CHEO: chrysanthemum essential oil; GEO: garlic essential oil; NLGEO: garlic essential oil nanoliposomes; CEO: cinnamon essential oil; PEO: polyethylene oxide.

**Table 1 foods-11-03017-t001:** Methods of liposomes preparation [2,11,25,26,27,28,29].

Method		Advantages	Disadvantages	Type of Vesicles
Conventional method	Thin film hydration	Simple process.	Low EE; organic solvent residue; small-scale production.	MLVs, GUVs
	Reverse phase evaporation	Simple process; suitable EE.	Organic solvent residue; time-consuming.	MLVs, LUVs
	Solvent Injection	Simple, rapid, and reproducible process.	Organic solvent residue; time-consuming; possible nozzle blockage (ether system).	SMVs, SUVs
	Detergent removal	Good particle size control; simple process.	Organic solvent and detergent residue; time-consuming; poor EE.	MLVs, LUVs
	Emulsion method	Simple process.	Low yield; organic solvent residue.	MVVs
	Heating method	Simple and fast process; no organic solvent; no sterilization is needed.	Degradation of bioactive compounds.	MLVs, SUVs
Advanced method	Cross-flow filtration	Rapid, scalable, sterile process; homogeneous size with high stability; easy removal of detergent.	Understudy method	SUVs, LUVs
	Modified ethanol injection	Simple, rapid, scalable, and continuous process; homogenous liposomes.	Organic solvent residue; high-cost material.	SUVs, LUVs
	Dual asymmetric centrifugation	Simple, rapid, and reproducible process; homogeneous and small liposomes; high EE for hydrophilic compounds.	Only laboratory-scale; high pressure with agitation.	SUVs, LUVs
	Microfluidic method	Good particle size control; scalable process and used for biological samples	Organic solvent residue; high cost and complex equipment.	SUVs, LUVs, GUVs
	Supercritical fluids	Control of particle size, possible in situ sterilization, low organic solvent consumption	High cost, high pressure, usage of sophisticated instruments.	LUVs

SUVs: small unilamellar vesicles; LUVs: large unilamellar vesicles; GUVs: giant unilamellar vesicles; SMVs: small multilamellar vesicles; MLVs: multilamellar vesicles; MVVs: multivesicular vesicles; EE: encapsulation efficiency.

**Table 2 foods-11-03017-t002:** Applications of encapsulated compounds in the meat industry.

Encapsulated Compounds		Meat/Meat Products	Effects		References
			Antimicrobial	Antioxidant	
Essential oils	Nutmeg essential oil	Pork, chicken, beef, mutton	Inhibit the growth of microorganisms (*E. coli* and *L. monocytogenes*) Have long-term acting antibacterial effect	/	[63]
		Dumplings	Improve the antibacterial effect on *L. monocytogenes* in dumplings. Extend the treatment time.	/	[64]
		Pork meat batters	Inhibit the growth of microorganisms (total viable counts)	Inhibit oxidation and decomposition of lipid and proteins (TBA, TVB-N, and carbonyl content)	[65]
	Thyme essential oil	Chicken	Improve the antibacterial effect on *S. enteritidis* Extend the treatment time.	/	[37]
	*Zataria multiflora* Boiss. essential oil	Beef burger	Inhibit the growth of microorganisms (total mesophilic and psychrotrophic bacteria, molds/yeast)	Inhibit oxidation and decomposition of lipid and proteins (peroxide, TVB-N)	[66]
Bacteriophages	Bacteriophage	Pork	Improve the antibacterial activity against *E. coli* O157:H7 in pork	/	[67]
		Beef	Inhibit *E. coli* O157:H7 growth in beef	/	[68]
Bioactive compounds	*Laurus nobilis* leaf extract	Minced beef	Inhibit the growth of microorganisms (total viable counts and psychrotrophic count, *E. coli* and *S. aureus*)	Inhibit oxidation and decomposition of lipid and proteins (peroxide and TBA value, free fatty acid value, TVB-N)	[69]
	Lupulon–xanthohumol	Cooked beef sausage	Inhibit the growth of microorganisms (total viable counts, *Clostridium perfringens*, coliforms, and molds/yeast) (nitrite + nanoliposome combination presented the best results).	Addition of liposome + nitrite successfully prevented lipid oxidation (TBARS)	[70,71]
	Catechin	Chinese dried pork	Inhibit the growth of microorganisms (total viable counts)	Inhibit lipid oxidation (peroxide, TBARS, pH value)	[38]
		Sauce duck	Inhibit the growth of microorganisms (total viable counts)	Inhibit oxidation and decomposition of proteins (TVB-N, pH value)	[72]
		Traditional Chinese bacon	/	Reduce the nitrosamines contents in fried bacon	[73]
Peptides	Quinoa peptide	Burger	Inhibit the growth of microorganisms (total viable counts, *S. aureus,* and molds/yeast)	Inhibit oxidation and decomposition of lipid and proteins (peroxide, TBARS value, TVB-N)	[74]

TBA: thiobarbituric acid; TVB-N: total volatile basic nitrogen; TBARS: thiobarbituric acid reactive substance.

**Table 3 foods-11-03017-t003:** Applications of biopolymer-liposome hybrid systems in meat products.

Biopolymer	Loaded Compounds	Meat/Meat Products	Effects	References
Chitosan coating	Laurel essential oils + nanosilver	Pork	Protected the quality of pork at 4 °C for 15 days	[36]
	*Satureja* plant essential oil	Lamb meat	Effectively retarded microbial growth and chemical spoilage	[83]
Chitosan and pectin coating	Chrysanthemum essential oil	Chicken	Showed high antibacterial activity against *C. jejuni* on chicken, while did not affect its quality	[41]
Chitosan films	Garlic essential oil	Chicken breast fillet	Showed significant synergistic effects in chemical and bacterial preservation of chicken fillet samples	[84]
Chitosan and whey protein films	Garlic essential oil	Sausage	Retarded lipid oxidation and the growth of main spoilage bacterial groups	[85]
PEO nanofibers	SiO_2_-eugenol	Beef	Higher antioxidant activity on beef	[86]
PEO/soybean lecithin-based nanofibers	Basil essential oil	Chilled pork	Help maintain the quality of chilled pork during 4-day storage	[43]
Chitosan/PEO nanofibers	Tea tree oil	Chicken	High antibacterial activity against *Salmonella*	[87]
Gelatin/chitosan nanofiber/ZnO nanoparticles nanocomposite film	Betanin	Beef	The growth of inoculated bacteria, lipid oxidation, and the changes in the pH and color quality of the beef samples were controlled by packaging with the fabricated film	[88]
CEO/β-CD proteoliposomes nanofibers	Cinnamon essential oil	Beef	The satisfactory antibacterial efficiency against *B. cereus* on beef was achieved without any impact on sensory quality of beef	[44]

PEO: polyethylene oxide; CEO: cinnamon essential oil.

## Data Availability

Not applicable.

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
