# Peer review of "Liposomes as Delivery System for Applications in Meat Products"

_foods, 2022, doi:10.3390/foods11193017_

Round 1

Reviewer 1 Report

The review entitled “Liposomes as delivery system for applications in meat products” has an interesting approach. Authors gave detailed information of theoretical aspects on liposomes and their application in meat products. In addition, the authors emphasize future perspectives on this matter. At the same time, the references employed for the elaboration of this review are generally adequate and it has many recent references.

Line 82-87: Why do you write the names of phospholipids with the first letter capitalized (for example, 1,2-Dis-tearoyl-82 sn-glycero-3-phosphocholine; and 1,2-Dipalmitoyl-sn-glyc-ero-3-phosphocholine? Consider using lowercase.

Line 95: In vitro and in vivo instead of in-vitro and in-vivo?

Line 96: Add a space after [24].

Line 118-119: Consider indicating in the figure caption the meaning of SUV, LUV, MLV, MVV. Although they are indicated in the text, they can make it easier for the reader to understand the image.

Table 1: Information is not provided very clearly. If possible, add horizontal lines to establish the corresponding divisions to make it easier for the reader to understand. The "size reduction technique" method is not actually a method for the preparation of liposomes, but rather it is an operation after their preparation (as is clearly stated in the text). Consider deleting the information regarding this method from the table, since the initial column refers to the preparation method and not to subsequent operations. Also, consider indicating in a table footer the meaning of the abbreviations (EE, MLV, GUV, etc.).

Line 201: Removes the extra space after "centrifugation,"

Line 201: Removes the extra space after "channel."

Line 252: It is the first time that you mention "L. monocytogenes" in the text. You should fully indicate the genus and species. It acts accordingly for the rest of the microorganisms (including tables if necessary).

Line 252: Eliminates (B. cereus), no need to specify

Line 252: It is the same title as line 239, so it should be removed and integrated into a single section.

Line 269: Replace "the" with "their".

Line 272: Removes the extra space after "preservation.".

Line 278: Removes the extra space after "losses.".

Line 96: Add a space after (iiiii).

Table 2: Includes a table footer with the meaning of the acronyms used (TBA, TVB-N, and TBARs). Replace "PH" with "pH".

Line 324: Add the word "and" before "Zataria"

Line 364: Shelf life or shelf-life? Be consistent throughout the manuscript.

Line 376: 24 d?

Lines 385-386: It is the first time that you include the acronyms TBARs and TVB-N in the text. Add its meaning.

Table 3: Includes a table footer with the meaning of the acronyms used (CEO and PEO).

Figure 6: Add the meaning of the acronyms in the figure caption.

Line 446: Removes the extra space after "beef.".

Author Response

Response to Reviewer 1 Comments

Point 1: Line 82-87: Why do you write the names of phospholipids with the first letter capitalized (for example, 1,2-Dis-tearoyl-82 sn-glycero-3-phosphocholine; and 1,2-Dipalmitoyl-sn-glyc-ero-3-phosphocholine? Consider using lowercase.

Response 1: Thank you for your kind suggestion! We have modified these mistakes in the revised manuscript (page 2, line72-73).

Point 2: Line 95: In vitro and in vivo instead of in-vitro and in-vivo?

Response 2: Thank you for your kind suggestion! We have modified the words in the revised manuscript (page 2, line 80).

Point 3: Line 96: Add a space after [24].

Response 3: Thank you for your kind suggestion! We added a space in the revised manuscript (page 2, line 84).

Point 4: Line 118-119: Consider indicating in the figure caption the meaning of SUV, LUV, MLV, MVV. Although they are indicated in the text, they can make it easier for the reader to understand the image.

Response 4: Thank you for your kind suggestion! We have indicated the meaning of SUV, LUV, MLV, MVV in the Figure 2 of the revised manuscript (page 4, line 142-143).

Point 5: Table 1: Information is not provided very clearly. If possible, add horizontal lines to establish the corresponding divisions to make it easier for the reader to understand. The "size reduction technique" method is not actually a method for the preparation of liposomes, but rather it is an operation after their preparation (as is clearly stated in the text). Consider deleting the information regarding this method from the table, since the initial column refers to the preparation method and not to subsequent operations. Also, consider indicating in a table footer the meaning of the abbreviations (EE, MLV, GUV, etc.).

Response 5: Thank you for your kind suggestion! We have deleted the information regarding the "size reduction technique" method from the table, and added a table footer (Table 1) in the revised manuscript (page 5, line 151-152).

Point 6: Line 201: Removes the extra space after "centrifugation,"

Response 6: Thank you for your kind suggestion! The extra space has been removed in the revised manuscript (page 7, line 233).

Point 7: Line 201: Removes the extra space after "channel."

Response 7: Thank you for your kind suggestion! The extra space has been removed in the revised manuscript (page 7, line 239).

Point 8: Line 252: It is the first time that you mention “L. monocytogenes” in the text. You should fully indicate the genus and species. It acts accordingly for the rest of the microorganisms (including tables if necessary).

Response 8: Thank you for your kind suggestion! We have modified it in the revised manuscript (page 8, line 285).

Point 9: Line 252: Eliminates (B. cereus), no need to specify

Response 9: Thank you for your kind suggestion! We have eliminated (B. cereus) in the revised manuscript (page 8, line 286)

Point 10: Line 252: It is the same title as line 239, so it should be removed and integrated into a single section.

Response 10: Thank you for your kind suggestion! We removed the title, and integrated into a single section “4.1. The release and bioavailability of delivered molecules” (page 8, line 271).

Point 11: Line 269: Replace "the" with "their".

Response 11: Thank you for your kind suggestion! “The” has been replaced with “their” in the revised manuscript (page 9, line 302).

Point 12: Line 272: Removes the extra space after "preservation."

Response 12: Thank you for your kind suggestion! The extra space has been removed in the revised manuscript (page 9, line 305).

Point 13: Line 278: Removes the extra space after "losses."

Response 13: Thank you for your kind suggestion! The extra space has been removed in the revised manuscript (page 9, line 311).

Point 14: Line 96: Add a space after (iiiii).

Response 14: Thank you for your kind suggestion! The extra space has been added in the revised manuscript (page 10, line 340).

Point 15: Table 2: Includes a table footer with the meaning of the acronyms used (TBA, TVB-N, and TBARs). Replace "PH" with "pH".

Response 15: Thank you for your kind suggestion! We have replaced "PH" with "pH" and added a table footer in the revised manuscript (page 12, line 356).

Point 16: Line 324: Add the word "and" before "Zataria"

Response 16: Thank you for your kind suggestion! The word “and” has been added in the revised manuscript (page 12, line 363).

Point 17: Line 364: Shelf life or shelf-life? Be consistent throughout the manuscript.

Response 17: Thank you for your kind suggestion! We have replaced “shelf-life” with “shelf life” throughout the whole revised manuscript (page 13, line 403).

Point 18: Line 376: 24 d?

Response 18: Sorry for making you puzzled. In the references [72], Song et al. treated duck with catechin nanoliposomes, showed a good fresh-keeping effect after 24 days of storage. We have corrected “24 d” with “24 days” in the revised manuscript (page 13, line 415).

Point 19: Lines 385-386: It is the first time that you include the acronyms TBARs and TVB-N in the text. Add its meaning.

Response 19: Thank you for your kind suggestion! Since the table footer (Table 2) has explained the meaning of the acronyms used in detail in the revised manuscript., we will not repeat it here (page 14, line 428-429).

Point 20: Table 3: Includes a table footer with the meaning of the acronyms used (CEO and PEO).

Response 20: Thank you for your kind suggestion! We have added a table footer (Table 3) in the revised manuscript (page 14, line 441).

Point 21: Figure 6: Add the meaning of the acronyms in the figure caption.

Response 21: Thank you for your kind suggestion! We have indicated the meaning of the acronyms in the Figure 6 in the revised manuscript (page 16, line 480-482).

Point 22: Line 446: Removes the extra space after "beef."

Response 22: Thank you for your kind suggestion! The extra space has been removed in the revised manuscript (page 16, line 494).

Reviewer 2 Report

L48- Please add "many research and reviews"

In section 2, there is too much of too general information is given, I recommend authors to remove subheadings and concise all information in to just 2 paragraphs and followed by the figures in one page.

Table 1 - contains of too much abbreviation and although, the authors have included explanation for abbrev before in the text, but it is mandatory to provide such information again in the note section below the table.

Table 1 is also lack of information regarding, what was the role of such liposome in that paper, did they use as delivery system, and what function there were intended to use for? please report

L175 - I recommend authors to provide more details regarding the size reduction and what the intention of doing it and how it is effectively help the food process, please include. 

Lack of mechanism is missing in Section 4.2, applicable to all subsections, for example, author mentioned about the liposome and their functions in each subheadings, but the given information is too general, and author should provide at least one mode of action of that functionality in the review. For example, author mentioned, liposome controls peptide functional loss under harsh environment, but where is the proof? just reference? I recommend authors to provide one or two lines of how do they achieve.

Figure 3-6 are from sources or from authors own creation? if author collect idea from some papers, please include reference and mention that you gather information from that paper etc.

Please remove Figure 7 from the review, as the conclusion generally doesn't contain any figures.

Author Response

Response to Reviewer 2 Comments

Point 1: L48- Please add "many research and reviews"

Response 1: Thank you for your kind suggestion! We have added "many researches and reviews" in the revised manuscript (page 2, line 50).

Point 2: In section 2, there is too much of too general information is given, I recommend authors to remove subheadings and concise all information in to just 2 paragraphs and followed by the figures in one page.

Response 2: Thank you for your thoughtful remarks! We have removed subheading and concise all information to just 2 paragraphs in the revised manuscript (page 2, line 61-139).

Point 3: Table 1 contains of too much abbreviation and although, the authors have included explanation for abbrev before in the text, but it is mandatory to provide such information again in the note section below the table.

Response 3: Thank you for your kind advice! We have added the meaning of abbreviation in the note section below the Table 1 (page 5, line 151-152).

Point 4: Table 1 is also lack of information regarding, what was the role of such liposome in that paper, did they use as delivery system, and what function there were intended to use for? please report.

Response 4: We appreciate the reviewer’s comments. Liposomes as delivery system have various kinds of functions in different applications. It seems hard to summarize their general functions in Table 1. In fact, Table 1 is about the preparation methods of liposomes, so it does not involve the role of such liposome. But as delivery system, we mentioned in 4.1 section that liposomes can control the release rate of encapsulated compounds to achieve long term action effect and increase the solubility of encapsulated compounds to improve bioavailability; 4.2 section mentioned that liposome can improve the stability of encapsulated compounds and play a better antibacterial and antioxidant effects. The above contents are their functions intended to use (page 8, line 271-299; page 11, line 345).

Point 5: L175 - I recommend authors to provide more details regarding the size reduction and what the intention of doing it and how it is effectively help the food process, please include.

Response 5: We thank the reviewer’s thoughtful remarks. Based on the suggestions of Reviewer 1 and you for this part, after our careful consideration, we decided to delete the information regarding the "size reduction technique" from the able 1, since it is not actually a method for the preparation of liposomes, but rather an operation after their preparation. We put the content of "size reduction technique" in the end of “section 3.1 Conventional methods”. According to your advice, we have provided more details regarding the size reduction and stated the intention of doing it in this part (page 6, line 204-210).

Point 6: Lack of mechanism is missing in Section 4.2, applicable to all subsections, for example, author mentioned about the liposome and their functions in each subheadings, but the given information is too general, and author should provide at least one mode of action of that functionality in the review. For example, author mentioned, liposome controls peptide functional loss under harsh environment, but where is the proof? just reference? I recommend authors to provide one or two lines of how do they achieve.

Response 6: Thank you for your kind suggestion! According to your suggestion, we have supplemented the relevant content and provided the corresponding evidence in Section 4.2. Researchers have investigated the decomposition temperature, degradation rate and activity retention rate of the encapsulated components exposed to the external environment, to confirm liposome can protect and stabilize the encapsulated compounds as protective barrier against environmental factors, in the references Pan, et al. [59]; Chen, et al. [60]; Folmer Correa, et al. [79]; Taylor, et al. [80] (page 11, line 346-351, page 13, line 420-424).

Point 7: Figure 3-6 are from sources or from authors own creation? if author collect idea from some papers, please include reference and mention that you gather information from that paper etc.

Response 7: We thank the reviewer’s helpful suggestion. The references about Figure 3 and Figure 6 have been mentioned in the revised manuscript (page 8, line 269-270, page 16, line 476,480).

Point 8: Please remove Figure 7 from the review, as the conclusion generally doesn't contain any figures.

Response 8: Thank you for your kind suggestion! We have removed Figure 7 in the revised manuscript.